# A First Report of *Sclerotinia sclerotiorum* Causing Forsythia Twig Blight in Romania

**DOI:** 10.3390/plants12203516

**Published:** 2023-10-10

**Authors:** Andreea-Mihaela Florea, Andrei-Mihai Gafencu, Florin-Daniel Lipșa, Iulian Gabur, Eugen Ulea

**Affiliations:** 1Department of Plant Science, Faculty of Agriculture, “Ion Ionescu de la Brad” Iasi University of Life Sciences, 3 Mihail Sadoveanu Alley, 700489 Iasi, Romania; amflorea@uaiasi.ro (A.-M.F.); agafencu@uaiasi.ro (A.-M.G.); 2Department of Food Technologies, Faculty of Agriculture, “Ion Ionescu de la Brad” Iasi University of Life Sciences, 3 Mihail Sadoveanu Alley, 700489 Iasi, Romania; flipsa@uaiasi.ro

**Keywords:** *Sclerotinia sclerotiorum*, new host, *Forsythia* spp., Sanger sequencing, ornamental plants

## Abstract

*Sclerotinia sclerotiorum* (Lib.) de Bary (1884) is a fungal plant pathogen with worldwide distribution and a varying host range from different botanical families. It can cause damage to a large variety of crops such as sunflower, soybean, dry bean, canola, some vegetables, and ornamental plants. This article reports the occurrence of twig blight on the forsythia plant from the NE region of Romania. The disease was observed on *Forsythia* × *intermedia Zab*. plants from the Arboretum Park of the Iasi University of Life Sciences (IULS), located in Iasi City, Romania. Infected tissue was investigated through morphological characteristics using Sanger sequencing. Genomic DNA was extracted from the isolate obtained from naturally infected plants, and the ribosomal internal transcribed spacer region was amplified using the ITS1, ITS2, and LSU D1 and D2. Based on the results of this study, molecular and morphological data suggest that Forsythia twig blight can be caused by *S. sclerotiorum*. Constant monitoring of *Sclerotinia sclerotiorum* across multiple hosts and time intervals will reduce potential spread and future economic losses in cultivated species.

## 1. Introduction

*Sclerotinia sclerotiorum* (Lib.) de Bary (1884) is a fungal pathogen that causes substantial losses to a wide range of hosts worldwide. *Sclerotinia sclerotiorum* (SS) is capable of infecting many plant species from different botanical families and can cause damage to a large variety of crops, such as sunflower, soybean, dry bean, canola, some vegetables, and ornamental plants [1]. Often, yield losses due to *S. sclerotiorum* infection exceed 20–35%, but the literature describes more documented cases of losses over 50% and up to 80–100% in temperate climates [2]. *S. sclerotiorum* has no specificity for a particular host and is able to infect monocotyledonous and dicotyledonous plant species [2,3]. Recent studies have found that *S. sclerotiorum* can also grow endophytically in monocots, including rice, wheat, maize, barley, and oat [4,5,6]. Based on infected host plant species, bioclimatic data, and geographic locations, economic losses caused by *S. sclerotiorum* are substantial and can vary extremely [7,8].

In Romania, *Sclerotinia sclerotiorum* is known for causing significant damage to plants in field crops and forced crops in greenhouses, warehouses, and silos [9]. The number of *Sclerotinia sclerotiorum* plant hosts at the national level is certainly higher [10], but reports on the presence of the fungus *Sclerotinia sclerotiorum* on woody ornamental plants are extremely scarce. Although *Sclerotinia sclerotiorum* is considered the main pathogen for a large number of herbaceous and woody ornamental species, only some information on the confirmed infection process and pathogenesis is available for *Sclerotinia sclerotiorum* in woody ornamental plants. In the scientific literature, there are numerous examples of ornamental plant infections; however, most articles do not include a description of the symptoms or severity of the infection [11,12].

More than 60 different *Sclerotinia sclerotiorum* disease symptoms have been reported in the literature as related to a broad host range that comprises agronomically important plant species [1]. *S. sclerotiorum* can infect multiple plant organs and survive in vegetable rests and soil for a long period, as sclerotia [13]. Sclerotia can infect new hosts in asexual form, as mycelia, or self-fertilized sexual reproduction, as ascospores, providing *S. sclerotiorum* with the possibility of accessing a wide range of potential targets [14].

The *Forsythia* genus is a group of plants in the olive family (*Oleaceae*) with around 11 species that are primarily native to eastern Asia, with one species from Europe. In Romania, *Forsythia* spp. Vahl is a perennial plant, appreciated and well known only as an ornamental plant named “golden-bells”. The same plant variety is also used for a wide range of Chinese medicines and health diets [15].

The majority of reports of *S. sclerotiorum* in ornamental plants come from the commercial production of plants in greenhouses or nurseries [16]. Therefore, less is known about the epidemiology of *S. sclerotiorum* in ornamental plants grown in natural landscapes [17]. In flowering woody ornamentals, descriptions of disease indicate that ascospores initiate infection through flowers. In *Forsythia* spp., plant water-soaked lesions form on flower petals in cool and moist weather conditions that are favorable for carpogenic germination. Afterward, Forsythia spp. infections progress into branch tissue, resulting in girdling cankers and the wilting of individual branches [18]. Moreover, in 2023, Munoz et al. [19] described *S. sclerotiorum* as the cause of rose bent neck on commercial cut roses (*Rosa hybrida* L.), while Rahman et al. (2020) [20] observed flower and blossom blight due to white mold disease on garden roses (*Rosa chinensis*) in Bangladesh. *Sclerotinia sclerotiorum* (Lib.) de Bary infection of ornamental plants of *Forsythia* spp. Vahl was initially suggested in samples collected in spring 2021 from the Arboretum Park of the Iasi University of Life Sciences (IULS), located in Iasi City, Romania [21].

This article describes the first confirmed *Sclerotinia sclerotiorum* infection on ornamental plants of *Forsythia* × *intermedia* Zab. (*F. suspensa* × *F. viridissima*) tissue [22] through morphological characteristics and genetic investigation using PCR and Sanger sequencing. Genomic DNA was amplified using ribosomal internal transcribed spacer region-specific primers, and samples were analyzed using sequencing techniques.

## 2. Results

### 2.1. Fungal Infection and Symptoms

In the spring of 2022, environmental conditions were extremely favorable for *Sclerotinia sclerotioum* (Lib.) de Bary initiation and evolution on *Forsythia* × *intermedia* Zab. ornamental plants from the Arboretum Park of the Iasi University of Life Sciences (IULS), located in Iasi, Romania (Figure 1). 

*S. sclerotioum* is a monocyclic pathogen, so there is only one primary mode of infection, depending on the type of infection incited [23]. Climatic conditions and plant vegetative growth stages have a major role in the infection spreading through ascospores. The floral elements, especially the stamens and the pistil, are highly sensitive to infection by ascospores. The ascospores completely colonized the mature and senescent flowers, and after a few days, the mycelium that colonized these flowers caused further infections through direct contact with the leaf and stem [24]. Water-soaked lesions progressed into branch tissue, resulting in the wilting of individual branches (Figure 2a). Necrotic tissues from the affected branches were examined, and we observed that they were covered with patches of fluffy white mycelia and sclerotia (Figure 2b).

To investigate the presence of *Sclerotinia sclerotiorum*, the fungus was isolated from infected tissues of Forsythia and cultured on a PDA medium. The fungus started to produce white masses of mycelium that grew to the edge of the Petri dish (Figure 3a). The size of the mycelium masses became bigger, and their colors became darker as time progressed. Furthermore, we observed that many black sclerotia formed on the Petri dish (Figure 3a). Under microscopic analysis, *Sclerotinia sclerotiorum* showed hyaline hyphae, branched and multinucleate (Figure 3b).

Four days after the pathogenicity assay, inoculated branches initially displayed discoloration, followed by water-soaked lesions that progressively developed at a distance of 5–10 cm from the contact point. Then, in a short time, the inoculated branches completely withered.

### 2.2. Sequence Alignment and Phylogenetic Analysis/Molecular Characterization

Genomic DNA isolated from plant material was subjected to PCR amplification of the 5.8S rRNA region, followed by Sanger sequencing. The PCR amplicon sizes obtained from the selected primes, ITS1/ITS2 and LSU D1/D2, were 580 bp. Information regarding the similarity of the investigated sequence was obtained by conducting BLAST searches (*blastn*) [25] against the GenBank database [26]. Genetic distance was investigated using simple neighbor-joining clustering based on uncorrected p-distances. BLAST results indicated that the nucleotide sequence on the selected sample had 99.91% similarity with the *Sclerotinia sclerotiorum* chromosome 7 sequence (Sequence ID: CP017820.1) and the *Sclerotinia sclerotiorum* isolate KR1121_1 (Sequence ID: KC311494.1). The phylogenetic analysis revealed that isolate SS_R was phylogenetically identical to other *S. sclerotiorum* isolates available in the GenBank database (Figure 4 and Appendix A).

A set of ten publicly available *S. sclerotiorum* sequences from different locations (accession numbers: UNIJSG.PL.OP1146, ATCC46762, KR1121_1, WU43983, KC311494.1), together with uncultured fungal genomic DNA sequences, were grouped together and formed a cluster. High similarities were observed among the SS_R isolate and other ascomycete fungi sequences obtained from leaf samples [19]. Phylogenetic analysis of the Forsythia twig blight sample with other fungal isolates from NCBI GeneBank indicated a high similarity of isolate SS_F to other *S. sclerotiorum* isolates. Based on the results of this study, molecular and morphological data suggest that Forsythia twig blight is caused by *Sclerotinia sclerotiorum*.

## 3. Discussion

The world’s cropping system has faced increasing challenges in recent years as climate change has advanced, creating optimal developmental conditions for fungal outbreaks, for example, white mold disease caused by *Sclerotinia sclerotiorum* [27]. The severe economic consequences of white mold infections can be addressed in major crops through integrated agro-management strategies. However, these strategies rely on a vast amount of fungicides that, in some cases, could trigger pathogen resistance to chemical compounds [28]. Moreover, increased quantities of chemical fungicides could also affect environmental and microbiota development at the plant level, suggesting that new disease management programs are necessary to avoid potential outbreaks or undesirable environmental, cultural, or biological side effects.

It is well known that *Sclerotinia sclerotiorum* is a typical necrophytic pathogen that produces severe economic losses, and recent studies showed that this fungus induces the production of reactive oxygen species (ROS), which leads to necrosis in the host, allowing the pathogen to absorb nutrients from the dead tissues. An important cofactor for enzymes scavenging ROS is represented by copper, which is an essential nutrient for microbial pathogens [29,30]. In fact, the metals most recognized for their crucial role in fungal homeostasis are elements like copper (Cu), iron (Fe), zinc (Zn), and manganese (Mn), which are extremely necessary for various biochemical processes, usually as enzyme cofactors. Other non-essential elements with an unknown biological role that can adversely affect microorganisms are cadmium (Cd), lead (Pb), and mercury (Hg) [31,32].

To our knowledge, this study represents the first report of *Sclerotinia sclerotiorum* infection in *Forsythia* × *intermedia* Zab. (*F. suspensa* × *F. viridissima*). Plant samples were collected from the Arboretum Park of the Iasi University of Life Sciences (IULS), and fungal infection was identified and validated through molecular analysis. The primary focus was given to the sclerotial mycoparasite morphological and molecular investigation and mode of action against *Forsythia* × *intermedia* Zab., an ornamental species frequently encountered in NE Romanian areas. Plant developmental stages, temperature, and humidity have an important role in the onset of *Sclerotinia sclerotiorum* infections [19]. In recent years, climatic changes like cool, wet weather in spring and variable winter conditions in the NE region of Romania have led to a more frequent appearance and increased incidence of *Sclerotinia sclerotiorum* on numerous ornamental plants.

Our understanding of the process of infection and pathogenesis for Sclerotinia-induced disease is lacking key components, while details regarding physiology, biochemistry, and molecular aspects are limited in the literature [33]. In this study, morphological analysis showed that the fungus *Sclerotinia sclerotiorum* infects the shoots of Forsythia only through the withering flowers, especially the stamens and the pistil, and extends up and down the woody branches from those points, causing wilting on the individual branches. Fungal-caused lesions progressed into branch tissue and led to a wilting effect. Taking into account the infection mechanism, effective management practices are necessary to limit the presence of this pathogen in other potential hosts and reduce the possibility of outbreaks.

Molecular identification of the isolated fungus, *Sclerotinia* sp., was further confirmed through PCR using ITS1, ITS2, and LSU D1 and D2 primers. PCR products produced amplicons of various sizes of about 586 bp, corresponding to the ITS and LSU regions. The amplicon was sequenced using Sanger. Munoz et al. observed similar results and reported the molecular characterization of *Sclerotinia sclerotiorum* using ITS 1 and ITS 4 on commercial cut roses (*Rosa hybrida* L.) [19].

Currently, twig blight symptoms are rare, and during this evaluation, we observed a relatively small area affected. However, constant monitoring of *Sclerotinia sclerotiorum* across multiple hosts and times will reduce the potential spread and future economic losses in cultivated species.

## 4. Materials and Methods

### 4.1. Diseased Sample Collection and Fungus Isolation

*Sclerotinia sclerotiorum* was observed on 18 May 2022 in *Forsythia* spp. plants from the Arboretum Park of the Iasi University of Life Sciences (IULS), located in Iasi, Romania (GPS coordinates: www.google.com/maps, accessed on 6 April 2023, 47°11′30.7″ N 27°33′25.2″ E). Infected tissues of the Forsythia host plant were collected and investigated in the research laboratory of the phytopathology discipline within the “Ion Ionescu de la Brad” Iasi University of Life Sciences (IULS). After a putative micromycete identification based on visual symptoms, fungal morphology, microscopic preparations, and a specialized guidebook, we followed a standard procedure [34] for fungal isolation to confirm the presence of *Sclerotinia sclerotiorum* in forsythia plants. Infected tissues of *Forsythia* × *intermedia* Zab. [22] were cut into small pieces and then rinsed 3~4 times with diluted water after being treated with 70% (*v*/*v*) ethanol for 2~3 s. The treated tissues were transferred to potato dextrose agar (PDA) medium and cultured at 25 °C. Then, whitish, cottony mycelium was transferred into PDA plates for 10 days to obtain pure cultures. White masses of mycelium developed to the edge of the Petri dish, and finally, many black sclerotia formed on the PDA medium of the selected isolate coded SS_F.

### 4.2. Pathogen Identification—Morphologic

Kock’s postulates were fulfilled using the isolate SS_F to inoculate six branches with a length of 25–30 cm of *Forsythia* × *intermedia* Zab. by placing a 1 cm agar plug with actively growing hyphal tips from a colony of *Sclerotinia sclerotiorum.* Six other branches were inoculated with an agar plug with no hyphae on the branch. All branches were maintained in a disinfected area with conditions for relative humidity close to 90% and a temperature of 28 °C with 10 h of light daily, after which water-soaked lesions similar to those in Arboretum Park samples were observed on the six inoculated branches.

### 4.3. Molecular Characterization of Sclerotinia Sclerotiorum Isolates

Genomic DNA was isolated using dried specimen samples of naturally infected plant material with the fungi. DNA was purified using the PCRBIO Rapid Extract Lysis Kit (https://pcrbio.com/row/products/dna-extraction/pcrbio-rapid-extract-lysis-kit/, accessed on 6 April 2023, PCR Biosystems Ltd., London, UK), following the manufacturer’s protocol. PCR amplification was conducted using the Platinum™ PCR SuperMix High Fidelity (https://www.thermofisher.com/order/catalog/product/12532016, accessed on 6 April 2023, Thermo Fisher Scientific, Waltham, MA, USA) and purification with ExoSap (https://www.thermofisher.com/order/catalog/product/78205.10.ML, accessed on 6 april 2023, Thermo Fisher Scientific, Waltham, MA, USA). The PCR reaction amplified four gene loci using the nuclear ribosomal internal transcribed spacer (ITS) and the large subunit ribosomal rRNA (LSU) as potential markers for species identification. The following primers were used: ITS1 (sequence: 5′- TCCGATGGTGAAC-CTGCGG-3′), ITS2 (sequence: 5′-ATGCGATACTTGGTGTGAAT-3′), and LSU D1 and D2 primers specific for the internal transcribed spacer region of the ribosomal DNA [35]. The ITS amplicon was amplified through an initial denaturation step at 5 min at 95 °C, 35 cycles of 30 s at 94 °C, 30 s at 55 °C, 40 s at 55 °C, and a final extension stage of 5 min at 72 °C. PCR products were verified using 1% agarose gel electrophoresis, comparing them with 2 Kb DNA markers. BMR Genomics (Padova, Italy) conducted genomic DNA extraction, PCR, and Sanger sequencing.

### 4.4. Sequencing and Phylogenetic Analysis

Sequencing was conducted using the BrilliantDye™ Terminator (v1.1) Cycle Sequencing Kit (1000 rxn) (https://www.nimagen.com/shop/products/brd1-1000/brilliantdye-terminator-v11-cycle-sequencing-kit-1000-rxn, accessed on 6 April 2023, NimaGen B.V., Nijmegen, The Netherlands) on a 3730xl DNA Analyzer (Thermo Fisher Scientific, Waltham, MA, USA). Results were submitted to rDNA homology sequence alignment using BLAST (https://blast.ncbi.nlm.nih.gov/Blast.cgi, accessed on 6 April 2023) and compared with the NCBI GenBank (GenBank; http://www.ncbi.nlm.nih.gov/BLAST/incidence.html, accessed on 6 April 2023) to verify the genetic similarity of the samples.

The phylogenetic tree was created using the NCBI-BLAST tool with the method “neighbor joining”, with a maximum seq difference of 0.5.

## 5. Conclusions

Awareness of favorable environmental conditions and knowledge of the life cycles of the host species can help to prevent *Sclerotinia sclerotiorum* infections. Familiarity with environmental conditions, disease dispersion methods, and continued scouting and sampling techniques should be included in an integrated management program in order to prevent *S. sclerotiorum* fungus incidence. Further investigations are needed for a better understanding of the germination and dormancy requirements of sclerotia caused by *S. sclerotiorum*. The inclusion of morphological, molecular, and climate data is crucial to the improvement of disease forecasting models.

## Figures and Tables

**Figure 1 plants-12-03516-f001:**
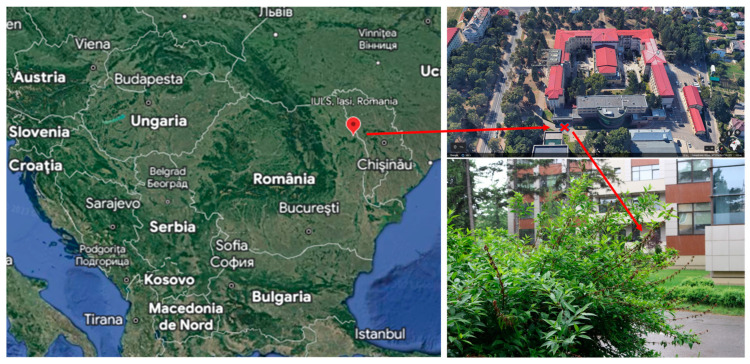
*Sclerotinia sclerotioum* on *Forsythia* × *intermedia* Zab. ornamental plant from the Arboretum Park of the Iasi University of Life Sciences (IULS), located in Iasi, Romania (GPS coordinates: www.google.com/maps, accessed on 6 April 2023, 47°11′30.7″ N 27°33′25.2″ E). Red arrows represent the exact location of the plant material investigated. Red x marks the physical location of the ornamental plant.

**Figure 2 plants-12-03516-f002:**
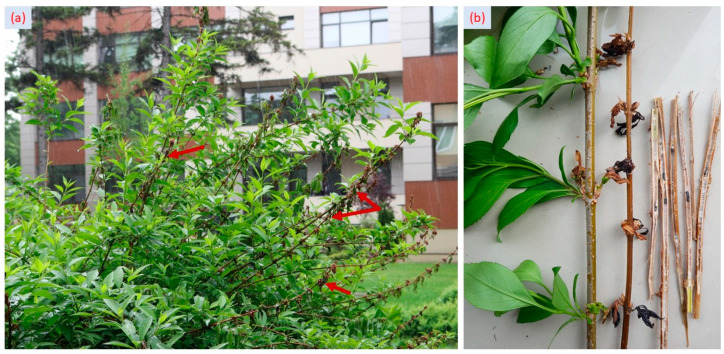
*Sclerotinia sclerotioum* on Forsythia. (**a**) Wilted branches of Forsythia attacked by *Sclerotinia sclerotioum*; (**b**) healthy tissues (**left** side) and necrotic tissues of Forsythia with patches of white mycelia and sclerotia (**right** side). Red arrows in (**a**) mark the sample extracted for further investigations and wilting of individual branches.

**Figure 3 plants-12-03516-f003:**
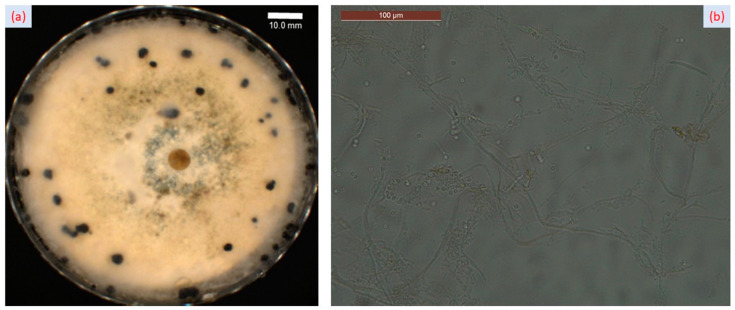
*Sclerotinia sclerotiorum.* (**a**) White masses of mycelium and sclerotia developed on PDA medium plate; (**b**) hyphae of *Sclerotinia sclerotiorum* at 400× magnification.

**Figure 4 plants-12-03516-f004:**
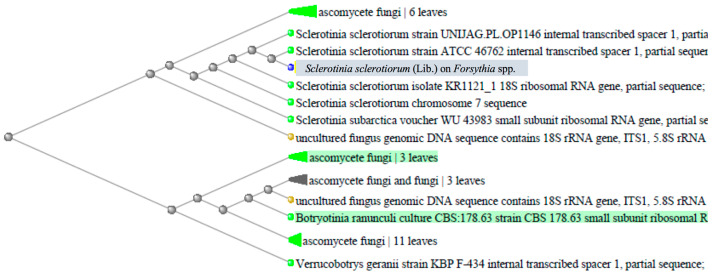
Genetic distance tree using the NCBI-BLAST tool, with “neighbor joining” method, max seq difference = 0.5. The sample analyzed in this paper is marked with grey. Sample sequence has a length of 580 bp and was amplified using PCR specific primers ITS1/ITS2 and LSU D1/D2.

## Data Availability

Data can be found in Appendix A.

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
