# Peer review of "A First Report of Sclerotinia sclerotiorum Causing Forsythia Twig Blight in Romania"

_plants, 2023, doi:10.3390/plants12203516_

Round 1
Reviewer 1 Report
Following your request to review this manuscript, this is my report which contains some remarks. In this paper entitled ' A first report of Sclerotinia sclerotiorum causing Forsythia twig 2 blight in Romania' .The authors described clerotinia sclerotiorum causing Forsythia twig 2 blight
The work is well done, carefully thought out, and performed and the manuscript is well written and falls within the scope of plants . To emphasize the MS for being suitable for consideration in this Jornal, I have a few comments as follows:
• Authors need to provide additional data on Sclerotinia sclerotiorum economic loses
• Authors need to provide the name of botanist of identified Forsythia spp. as well as s ID of identification
• Discussion should be extended by revising additional papers dealing with heavy metal Sclerotinia sclerotiorum presence worldwide
• Please update references to be within the last five years as much as possible since you included many old references
This MS merits publication after addressing minor points raised here
language is ok
Author Response
Thank you for the valuable comments and requests of the reviewers to improve the quality of our manuscript. As requested by the in-house editorial comments and by the reviewers to focus and clarify the main message of the manuscript we changed all parts of the manuscript and added additional information and references.
In the following sections, we address the questions of the reviewers (shown below in italics) one by one.
Comments to reviewer 1:
Following your request to review this manuscript, this is my report which contains some remarks. In this paper entitled ' A first report of Sclerotinia sclerotiorum causing Forsythia twig 2 blight in Romania' .The authors described clerotinia sclerotiorum causing Forsythia twig 2 blight
The work is well done, carefully thought out, and performed and the manuscript is well written and falls within the scope of plants . To emphasize the MS for being suitable for consideration in this Jornal, I have a few comments as follows:
- Authors need to provide additional data on Sclerotinia sclerotiorum economic loses
To focus more on the economic loses of S. sclerotiorum aspects we added new literature references and paragraphs in the introduction and the discussion as follows:
We changed parts of the Introduction to clarify that the major economical loses aspect of this paper (lines 31 to 38). Added a new paragraph in the Discussions (lines 223-225) and linked it with the other reviewers comments on heavy metals.
- Authors need to provide the name of botanist of identified Forsythia spp. as well as s ID of identification
We added a new literature reference in the Introduction (reference 22) and corrected the Forsythia spp. spelling throughout the manuscript.
- Discussion should be extended by revising additional papers dealing with heavy metal Sclerotinia sclerotiorumpresence worldwide
We added in the Discussion section a new paragraph that deals with heavy metal S. sclerotiorum presence (lines 223-232).
- Please update references to be within the last five years as much as possible since you included many old references
We agree to this point and rephrased large parts of the manuscript text by adding 12 new literature references published in the last five years.
This MS merits publication after addressing minor points raised here
Reviewer 2 Report
In this manuscript (plants-2632098) entitled "A first report of Sclerotinia sclerotiorum causing Forsythia twig blight in Romania" submitted to Plants, Andreea Mihaela Florea and colleagues the occurrence of twig blight on forsythia plant from the NE Region of Romania. This research is interesting, but the present manuscript is unsuitable for publication.
1. For Figure 1, detailed position of arboretum park of Iasi University of Life Sciences (IULS) on the map of Romania should be included in the revised Figure.
2. For Figure 1a, wilted branches of Forsythia spp. attacked by Sclerotinia sclerotioum should be indicated by arrows.
3. For Figure 1b, healthy tissues of Forsythia spp. without patches of white mycelia and sclerotia should be included as controls.
4. For Figure 2, where is scale bar?
5. For Figure 2, microscopic analysis of Sclerotinia sclerotioum development should included in the revision.
Author Response
Thank you for the valuable comments and requests of the reviewers to improve the quality of our manuscript. As requested by the in-house editorial comments and by the reviewers to focus and clarify the main message of the manuscript we changed all parts of the manuscript and added additional information and references.
In the following sections, we address the questions of the reviewers (shown below in italics) one by one.
Comments to reviewer 2:
In this manuscript (plants-2632098) entitled "A first report of Sclerotinia sclerotiorum causing Forsythia twig blight in Romania" submitted to Plants, Andreea Mihaela Florea and colleagues the occurrence of twig blight on forsythia plant from the NE Region of Romania.
- For Figure 1, detailed position of arboretum park of Iasi University of Life Sciences (IULS) on the map of Romania should be included in the revised Figure.
- For Figure 1a,wilted branches of Forsythia spp. attacked by Sclerotinia sclerotioumshould be indicated by arrows.
- For Figure 1b,healthy tissues of Forsythia spp. without patches of white mycelia and sclerotia should be included as controls.
Thank you very much for your comments. We revised Figure 1 form the initial manuscript and created two new figures 1 and 2.
- The new Figure 1 contains detailed position of arboretum park of Iasi University of Life Sciences (IULS), including GPS coordinates.
- The new updated Figure 2 contains (2a) wilted branches of Forsythia attacked by Sclerotinia sclerotioum, marked by arrows and (2b) healthy tissues (left side) and necrotic tissues of Forsythia with patches of white mycelia and sclerotia (right side)
- For Figure 2,where is scale bar?
- For Figure 2,microscopic analysis of Sclerotinia sclerotioumdevelopment should included in the revision.
We modified the original Figure 2 and added scale bar (Figure 3a,3b) and included microscopic analysis of the S. sclerotiorum development images (Figure 3b).